# Time Duration of Post-Activation Performance Enhancement (PAPE) in Elite Male Sprinters with Different Strength Levels

**DOI:** 10.3390/children10010053

**Published:** 2022-12-26

**Authors:** Wenxia Guo, Meifu Liang, Junlei Lin, Ruihang Zhou, Ningning Zhao, Felipe J. Aidar, Rafael Oliveira, Georgian Badicu

**Affiliations:** 1Department of Social Sports, Beijing University of Chemical Technology, Beijing 100029, China; 2Chinese Athletics Association, Beijing 100061, China; 3School of Strength and Conditioning, Beijing Sport University, Beijing 100084, China; 4China Institute of Sport Science, Beijing 100061, China; 5Sports Science Institute of Hebei Province, Shijiazhuang 050000, China; 6Graduate Program of Physical Education, Federal University of Sergipe (UFS), São Cristovão 49100-000, Brazil; 7Sport Science School of Rio Maior-Polytechnic Institute of Santarem, 2040-413 Rio Maior, Portugal; 8Life Quality Research Centre, 2040-413 Rio Maior, Portugal; 9The Research Centre in Sports Sciences, Health Sciences and Human Development (CIDESD), 5001-801 Vila Real, Portugal; 10Department of Physical Education and Special Motricity, Transilvania University of Brasov, 500068 Brasov, Romania

**Keywords:** strength training, PAPE, peak power, time course, back squat, static squat jump

## Abstract

(1) Purpose: This study aimed to explore the time duration of post-activation performance enhancement (PAPE) in elite male sprinters with different strength levels. (2) Methods: Thirteen elite male sprinters were divided into a strong group (relative strength: 1RM squat normalized by body mass of ≥2.5; *n* = 6) and a weak group (relative strength of <2.5; *n* = 7). All sprinters performed one static squat jump (SSJ) at baseline and 15 s, 3 min, 6 min, 9 min, and 12 min following an exercise protocol including three reps of a 90% 1RM back squat. Two force plates were used to determine the vertical jump height, the impulse output, and the power output for all SSJs. (3) Results: Significant improvements in vertical jump height and peak impulse were observed (*p* < 0.05) at 3, 6, and 9 min, without significant between-group differences. The peak power had a significant increase in 3 min (*p* < 0.01) and 6 min (*p* < 0.05), with also no significant difference between-group differences. Moreover, the stronger subjects induced a greater PAPE effect than the weaker counterparts at 3, 6, and 9 min after the intervention. The maximal benefit following the intervention occurred at 6 min and 3 min after the intervention in the stronger and weaker subjects, respectively. (4) Conclusions: The findings indicated that three reps of a 90% 1RM back squat augmented the subsequent explosive movement (SSJ) for 3–9 min in elite male sprinters, especially in stronger sprinters.

## 1. Introduction

There is an acute positive response of explosive performance to a conditioning activity (CA) [1,2]. This enhancement has recently been termed as post-activation performance enhancement (PAPE) [1,3,4,5]. The underlying mechanisms for PAPE have been primarily attributed to the following physiological effects: positive changes of muscle temperature, fiber water content, and muscle activation [1,5,6,7]. Although there is no evidence regarding the exact mechanisms, the acute benefits of PAPE have been observed in several explosive movements such as jumping [8,9], throwing [10,11,12], sprinting [13,14,15], and changing of the direction [16]. Therefore, the ability to harness PAPE effects to maximize athletic performance has received intensive attention [17,18,19].

As the CA used to induce PAPE can also induce fatigue [20,21], the optimal interval time post-CA required to induce PAPE is of great interest. The window of opportunity seems relatively brief [22], and the effect of PAPE dissipates over time. If the appropriate recovery is provided, the muscle will be in a potentiated state only [20]. Conversely, if there is insufficient rest, the muscle will be in a fatigued state and performance will be impaired [23]. Thus, the interaction of fatigue and potentiation should be taken into account [24], and it seems that the maximization of PAPE is largely determined by an optimal interval time. Based on previous studies examining the interval time from 0 to 21 min [25,26,27,28,29,30] and due to the differences between participants’ characteristics, there is no available consensus indicating the optimal PAPE “window” between a CA and the subsequent explosive activities. For example, Beato [31] reported that PAPE can induce a marked improvement in counter movement jump (CMJ) performance following 3–7 min rest intervals, which is in line with the findings of previous studies [32,33]. In addition, a single bout of eccentric overload resistance intervention produced comparable PAPE benefits but with a single bout of longer duration than a previous study [25]. However, several other studies have failed to find any augments in the participant’s lower body performance from 15 s to 20 min after performing sub-maximal back squats [34,35]. To date, few evidence investigated the time duration of PAPE using discontinuous strategies [18], and little research has determined the optimal PAPE time windows for the utilization of discontinuous strategies in elite male sprinters.

With respect to the athlete’s characteristics, the strength level is a dominant factor influencing the induction of PAPE. Evidence suggests that stronger players elicit a greater PAPE effect compared to weaker players [28,36]. A study by Gourgoulis et al. [8] reported stronger athletes (squat weights > 160 kg) had a greater increase in CMJ height (4% vs. 0.4%) compared to weaker athletes (squat weights < 160 kg) following five sets of back squats. Similarly, Seitz et al. [36] observed that stronger participants expressed a significantly greater PAPE response than weaker counterparts at all post-CA squat jump tests. A plausible explanation for this phenomenon might be related with the athlete fiber type distribution [20], and there is a positive strong correlation of the maximal strength with the ratio of type II muscle fibers [37,38,39]. Furthermore, muscles, with a greater ratio of type II muscle fibers, would have a larger amount of higher-order motor units that are more sensitive to PAPE mechanisms following a CA [40].

Moreover, little evidence examined the time duration of PAPE in elite male sprinters. Therefore, the primary purpose of this study was to examine the time windows for the maximal PAPE benefits between a CA (one set of three back squats at a 90% 1RM) and a lower body explosive activity in a group of elite male sprinter athletes. It was hypothesized that there would be a significant time duration between the CA and PAPE response in elite male sprinters. A secondary purpose was to verify the role of the conditioning activity on the performance in the back squat in runners with different levels of strength. We expected that stronger sprinters would exhibit a higher potential benefit than weaker sprinters did.

## 2. Materials and Methods

### 2.1. Participants

Thirteen elite male sprinters (Table 1), competing in 100 m races at the national or collegiate level in China, volunteered for this study. Inclusion criteria required squat practice experience of more than two years, no cardiovascular diseases, no caffeine intake in the previous 3 h, and no lower-limb or back injury in the previous 3 months. All subjects were recruited from a leading university in China and divided into a strong group and a weak group, depending on their relative 1RM squats. The strong group referred to the subjects with a 1RM back squat normalized by body mass of ≥2.5, and the weak group referred to the subjects with a 1RM back squat normalized by body mass of <2.5 [36]. Data collection was conducted at the Athletics Gymnasium. This study was approved by the ethical committee of China Institute of Sport Science (CISS). All subjects signed a written informed consent form before data collection.

### 2.2. Procedures

The current research asked the participants to complete one familiarization session and one experimental session. The familiarization session involved a 1RM back squat test and familiarization with the experimental process. All athletes performed 1RM back squat testing following the guidelines of the National Strength and Conditioning Association (NSCA) testing protocol [41]. Height and body mass were measured by a calibrated Xiangshan ultrasonic rangefinder (HT-01; Xiangshan Ltd., Beijing, China) and a Xiangshan electronic scale (EF866i; Xiangshan Ltd., Beijing, China), respectively.

During the experimental session, the participants performed a standardized 10 min warm-up strategy involving submaximal cycling, dynamic stretching, and vertical jump [42]. Following the warm-up, a 3 min recovery interval was provided prior to the measurements of baseline. After the warm-up, the participants performed a proper SSJ as high and as fast as possible [43,44]. Specifically, 5 min after the baseline SSJ data were determined, the participants then completed 3 repetitions at a 90% 1RM as the CA and single SSJs at 15 s and 3, 6, 9, and 12 min post-CA. Consistent verbal encouragement was provided to ensure consistency within and across all the attempts. This conditioning activity protocol was deemed appropriate based on previous recommendations [36,45], coordinated with the subjects’ current resistance training regimens and allowed for the lowest fatigue to be affected while allowing PAPE responses to generate [20]. The assessment times of the post-CA SSJs were determined ground on previous research exploring the time duration of PAPE [45,46]. All assessments were performed after the main meal at the same time of the day.

### 2.3. Data Collection and Analysis

All SSJ measurements were performed using a force platform (9281E; Kistler Group, Switzerland) with a commercial software (BioWare, version 5.3.0.7; Kistler, Group.) that allowed direct measurement of vertical ground reaction forces (VGRFs) and sampling at a frequency of 1000 Hz. Calibration was checked with a known mass before testing [47].

Vertical jump height was calculated from time-in-air, identified as the duration between the take-off and landing contact times utilizing the assumption of uniform acceleration. Vertical jump height was acquired via the following equation: vertical jump height = $g\times \frac{t^{2}}{8}$, where g is 9.81 m·s^−2^ and t is the time-in-air. The interclass correlation for vertical jump height following this procedure was 0.89. The instantaneous impulse was based on the VGRF and the time. The instantaneous power output was determined by the impulse-momentum approach based on the method of previous research [48]. The peak impulse and power were determined using the following basic equations: $v_{0}=0$, $F_{i}\times t_{i}=m\times \left(v_{n+1}-v_{n}\right)$, P = $F_{i}\times v_{i}$, I = $F_{i}\times t_{i}$, where *F* is the force, *t* is described as 1/sampling frequency, *v* is the velocity, *m* is the body mass, P is the power, and I is the impulse. The initial velocity of the movement was zero [48]. The interclass correlation values for the peak impulse and the power were 0.93 and 0.94, respectively. The jump performances between baseline and post-jump at each time interval was compared via the equation: % Difference = $\frac{\mathrm{post}-\mathrm{pre}}{\mathrm{pre}}\times 100$.

### 2.4. Statistical Analysis

The statistics were performed by SPSS version 22.0 (SPSS Inc., Chicago, IL, USA) for Windows. Mixed two-factors (2 × 6) repeated-measures analyses of variance (ANOVA) and least significant difference (LSD) pairwise comparisons with Bonferroni corrections were used to make comparisons within and between groups. The effect size was calculated (η_p_^2^), and values of <0.06 represented a small effect size, values of <0.14 represented a medium effect size, and values of >0.14 represented a large effect size [49]. Power analysis was conducted with G’Power version 3.1.9.6 software (Heinrich Heine University, Düsseldorf, Germany) to calculate the correct sample size. Considering an alpha error of 0.05 and a statistical power of 80%, a minimum of 12 participants was required to detect a small overall effect size (ES) of 0.32 [50,51]. Following a test for the normality of distribution, the results are expressed as mean ± *SD*. The significance level was set at *p* < 0.05.

## 3. Results

### 3.1. Vertical Jump Height

When comparing the six individual vertical jump heights of SSJs at different times with the mixed two-factors repeated-measures ANOVA, it was determined that there was a significant main effect for time (*F* = 8.503; η^2^p = 0.436; *p* < 0.01). The pairwise comparison indicated that the maximum vertical jump height was observed at 6 min (38.38 ± 5.6 cm vs. 42.04 ± 5.6 cm; *p* < 0.01; CI: −0.061 to −0.013; MD = −0.037), and was significantly higher than the vertical jump height recorded at baseline. In addition, the vertical jump height had a significant increase at 3 min (38.38 ± 5.6 cm vs. 41.65 ± 6.6 cm; *p* < 0.01; CI: −0.5 to −0.15; MD = −0.032) and 9 min (38.38 ± 5.6 cm vs. 41.70 ± 6.0 cm; *p* < 0.01; CI: −0.055 to −0.013; MD = −0.034) compared with their baseline. No significant differences were noted for the interaction effect between the time and the group (*F* = 1.197; η^2^p = 0.098; *p* > 0.05) or the main effect for group (*F* = 0.008; η^2^p = 0.001; *p* > 0.05) (Figure 1). The optimal recovery time to maximize PAPE effects on the vertical jump height was 3 to 9 min.

Mixed two-factors (2 × 6) repeated-measures analyses of variance was used to compare the vertical jump height within and between groups. * indicates the significant difference from baseline (*p* ≤ 0.05). ** indicates the significant difference from baseline (*p* ≤ 0.01).

As shown in Figure 2, the percentage difference from the baseline of the strong group demonstrated a smaller percent increase at 15 s (0.25% ± 8.19% vs. 2.61% ± 4.41%) after the CA and a greater percent increase at 3 min (8.26% ± 7.09% vs. 8.75% ± 7.65%), 6 min (12.80% ± 14.48% vs. 7.84% ± 5.28%), 9 min (12.18% ± 11.43% vs. 6.26% ± 6.59%), and 12 min (9.10% ± 13.81% vs. 3.05% ± 6.29%) compared with for the weak group.

### 3.2. Peak Impulse

The mixed two-factors repeated-measures ANOVA revealed a significant time effect on the peak impulse over the duration of the study (F = 3.840; η^2^p = 0.259; *p* < 0.01). The pairwise comparison indicated that the peak impulse had a significant increase at 3 min (172.07 ± 41.75 N·s vs. 185.17 ± 42.26 N·s; *p* < 0.05; CI: −24.97 to −1.54; MD = −13.26), 6 (172.07 ± 41.75 N·s vs. 187.46 ± 49.51 N·s; *p* < 0.05; CI: −28.92 to −2.83; MD = −15.87), and 9 min (172.07 ± 41.75 N·s vs. 184.39 ± 38.52 N·s; *p* < 0.05; CI: −23.48 to −2.00; MD = −12.74) in comparison to the baseline. No significant differences were noted for the interaction effect between the peak impulse and the group (F = 1.140; η^2^p = 0.094; *p* > 0.05) or the main effect for the group (F = 0.215; η^2^p = 0.019; *p* > 0.05) (Figure 3). The optimal recovery time to maximize PAPE effects on the peak impulse was 3 to 9 min.

Mixed two-factors (2 × 6) repeated-measures analyses of variance was used to compare the peak impulse within and between groups. * indicates the significant difference from baseline (*p* ≤ 0.05).

The strong group exhibited a smaller increase in peak impulse at 15 s than the weak group (2.31% ± 10.78% vs. 4.51% ± 5.70%). The percentage difference from baseline demonstrated a greater percent increase at 3 min (9.94% ± 14.84% vs. 7.29% ± 8.28%), 6 min (13.70% ± 16.09% vs. 5.19% ± 10.30%), 9 min (11.49% ± 14.48% vs. 5.73% ± 9.72%), and 12 min (4.39% ± 10.12% vs. 5.18% ± 10.72%) after the conditioning activity compared with for the weak group (Figure 4).

### 3.3. Peak Power

The mixed two-factors repeated-measures ANOVA revealed a significant time effect (*F* = 3.713; η^2^p = 0.252; *p* < 0.01). The pairwise comparison indicated that the peak power had a significant increase at 3 min (3894.18 ± 1206.12 W vs. 4269.21 ± 1186.33 W; *p* < 0.01; CI: −649.79 to −113.86; MD = −381.82) and 6 min (3894.18 ± 1206.12 W vs. 4324.95 ± 1479.81 W; *p* < 0.05; CI: −804.44 to −79.10; MD = −441.77) compared with their baseline. No significant differences were noted for the interaction effect between the peak power and the group (*F* = 1.308; η^2^p = 0.106; *p* > 0.05) or the main effect for group (*F* = 0.740; η^2^p = 0.063; *p* > 0.05) (Figure 5). The optimal recovery time to maximize PAPE effects on peak power was 3 to 6 min.

Mixed two-factors (2 × 6) repeated-measures analyses of variance was used to compare the peak power within and between groups. * indicates the significant difference from baseline (*p* ≤ 0.05). ** indicates the significant difference from baseline (*p* ≤ 0.01).

As shown in Figure 6, the strong group exhibited a smaller increase in peak power at 15 s (4.05% ± 10.89% vs. 7.74% ± 8.67%) and 12 min (−0.54% ± 8.86% vs. 8.09% ± 15.14%) than the weak group. In addition, the peak power had a greater increase at 3 min (12.52% ± 15.07% vs. 10.01% ± 9.60%), 6 min (14.95% ± 18.35% vs. 7.41% ± 12.84%), and 9 min (11.17% ± 19.37% vs. 8.84% ± 13.71%) post-CA compared with for the weak group (Figure 6).

## 4. Discussion

The present study aimed to investigate the optimal recovery time between male sprinters with different strength levels during an SSJ test providing three reps of a 90% 1RM squat. The findings of the current evidence suggest that the optimal recovery time to maximize PAPE benefits of lower body performance is 3 to 9 min in elite male sprinters. In addition, we confirmed our previous hypothesis, as the key finding of this study indicated that the strong group exhibited the maximum PAPE benefits for the vertical jump performance, the peak impulse, and the peak power at 6 min, whereas the weak group expressed the maximum PAPE effect at 3 min after intervention. Furthermore, the strong group performed greater PAPE effects on the vertical jump height, the peak impulse output, and the peak power output than the weak group at 3, 6, and 9 min after intervention.

The present study aimed to investigate the optimal interval time between the CA and the subsequent lower body performance on sprinters. Previous investigations have selected recovery intervals ranging from 0 to 21 min [26,27,28,29,30] for the lower body, with no consensus reached to date on the optimal time required. Young et al. [33] selected a 4-min rest period and demonstrated a 2.8% improvement on the CMJ performance. Evetovich et al. [30] reported that the vertical jump performance, the horizontal jump performance, and the sprint performance were significantly increased at 8 min post-CA. Bogdanis et al. [29] used various recovery intervals (15 s and 2, 4, 6, 8, 10, 12, 15, 18, and 21 min) and reported a significant increase between the CMJ performance at 2–10 min of recovery and the CMJ performance performed before the pre-load. Crewther et al. [52] reported that the CMJ height was significantly improved at 4, 8, and 12 min compared with the baseline values after 3RM squats. Most studies are in line with the results demonstrated by meta-analyses [53], which pointed out greater effects of the PAPE after 3–10 min of recovery post-CA than with rest intervals that were shorter than 2 min and longer than 16 min. However, these findings serve to elucidate the optimal recovery (3–9 min) required to reach the maximal PAPE effect in the lower body performance in elite male sprinters. This was consistent with the previous literature that the greatest PAPE effect was determined approximately 2–10 min [52,54] between the CA and the subsequent lower body explosive performance.

Notably, the temporal profile of PAPE for reaching the greatest PAPE benefits would be determined by the individual strength level [8,28,36,45]. This phenomenon has been evidenced by Kilduff et al. [45], who observed a correlation (*r* = 0.63; *p* < 0.01) between the subject’s strength (absolute and relative) and the CMJ peak power potentiation after a PAPE protocol. A previous study reported that stronger players elicited a 4% augment in CMJ height (*p* < 0.05) after a CA [8]. Conversely, weaker counterparts (squat loads <160 kg) only recorded a 0.4% increase in CMJ height (*p* > 0.05). In addition, Seitz et al. [36] found that significant greater PAPE responses were observed in the stronger group compared to in the weaker group. Sañudo et al. [4] recently confirmed that a superior PAPE benefit was gained in stronger individuals after performing a CA. Likewise, our research shows that sprinters with high strength levels can induce higher PAPE benefits compared to their weaker counterparts following intervention of a set of three squats with a 90% 1RM.

Seitz et al. [55] already showed that the strength of potentiation is determined by the strength level and training experience. In addition, evidence reported that stronger players performed their maximal squat jump height (*p* = 0.002; ES = 0.90) earlier than their weaker group (*p* = 0.01; ES = 0.56) after a PAPE protocol [36]. However, our results contradict with those of Seitz et al. [36], who found that sprinters with high strength levels can maximize PAPE benefits better than sprints with lower strength levels post-CA. Greater PAPE benefits were observed in stronger players. This may be because they tend to have larger and stronger type II muscle fibers and display elevated myosin light chain phosphorylation [20,24], which might increase their ability to harness PAPE. Considering the interaction of fatigue and potentiation, more time is required to exhibit the maximum PAPE response later in stronger sprinters than in weaker counterparts. The different time duration of PAPE might be due to the relationship between fatigue and potentiation [55]. Specially, enhanced activation of fast motor units also evokes a stronger PAPE response, which may also lead to larger fatigue. As a result, stronger sprinters may be able to dissipate fatigue slower than weaker counterparts after the high intensity resistance exercises because of their higher ratio of type II muscle fibers [21].

The main limitation of the study is a relatively small number of subjects were recruited in this study because of limited access to elite male sprinters. The restricted statistical power because of the sample size in this study may have influenced the significance of some of the statistical comparisons conducted. A post hoc power analysis revealed that, for the medium effect size of interest observed in the present study (ES *f* = 0.25), the number of players would have been at least 10 for each group to obtain statistical power at the recommended 80% level. The results of the present study serve as a basis that can be generalized for larger populations. Thus, more investigation with larger samples is needed to determine the effects of PAPE in male sprinters and related sports. Another limitation should be noted is that the PAPE benefits induced by the CA cannot be completely isolated in the present study because of the lack of a control group. Future studies should use a control group in their design.

## 5. Practical Applications

This study found that statistically significant and practical improvements in lower body performance were occurred in 3 to 9 min following three reps of a 90% 1RM squat in elite male sprinters. Thus, this strategy can be utilized in daily routine training programs, as well as warming-up sessions for competition. A large body of evidence demonstrated that positive adaptive responses of muscle and tendon to the long-term PAPE training protocol were observed in elite male and female athletes. For example, complex training [56,57], the mechanisms of which involve PAPE, is a mixed-training method that is high-load resistance training followed by low-load jumping training within a session. A previous study investigates the effect of complex training on explosive performance of upper and lower body in early pubertal boys, and the authors reported that a 12-week exercise intervention significantly improved selected explosive performance [58].

It is worth noting that the PAPE benefits induced by a CA were largely determined by the interaction of potential and fatigue. The maximal PAPE benefit, thus, is time-dependent. However, the optimal window is also determined by several factors, such as the strength level, gender, sports, and the intervention type. The optimal window reported in the present study may only be valid in elite male sprinters. We suggest that it is better for practitioners to determine their optimal windows.

## 6. Perspective

The present study indicated that PAPE benefits induced by three reps of a 90% 1RM squat occurred at 3 to 9 min following three reps of a 90% 1RM squat, and the maximal PAPE benefits were observed at 6 min and 3 min after the CA in the stronger and weaker subjects, respectively. These findings support that the optimal window of PAPE benefits was determined by the individual’s strength level and supplements the limited evidence in this filed, especially for elite male sprinters. However, as mentioned above, PAPE benefits and the optimal window may be determined by the intervention type. According to the training-specific hypothesis and the dynamic correspondence theory, the movement direction of the intervention may induce varied PAPE benefits in vertical-oriented and horizontal-oriented jump performance or force−velocity profiles. Thus, future studies should explore whether the force-vector theory can also work in PAPE.

## 7. Conclusions

The results of the present study found that one set of three back squats at a 90% 1RM providing 3–9 min recovery intervals did uniformly augment subsequent explosive movement in elite male sprinters in China. In addition, the temporal profile of PAPE was associated with the sprinter’s initial strength, as stronger sprinters exhibited a greater PAPE effect following a set of submaximal back squats. Additionally, it appeared that weaker sprinters exhibited their maximal PAPE responses earlier than the stronger counterparts. Finally, we recognized the limitations of the present study. Given the small sample size in this study, a bigger sample size in the future studies is needed to enhance the generalization of the research findings. Moreover, the printing performance still needs to be carried out to see if the PAPE can be harnessed between two different strength levels among elite male sprinters.

## Figures and Tables

**Figure 1 children-10-00053-f001:**
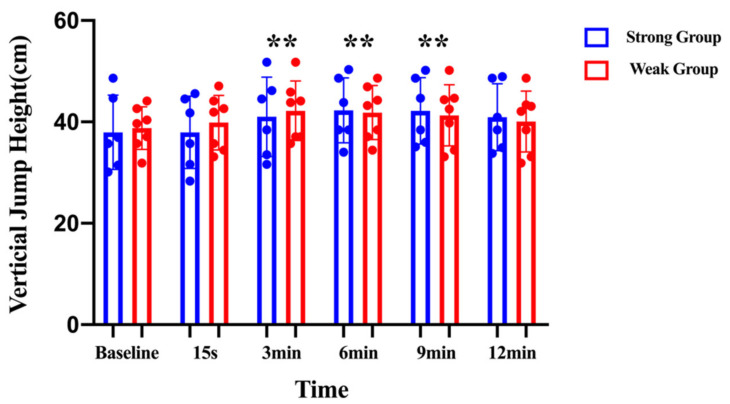
Vertical jump heights before and after the conditioning activity for the strong and weak groups. ** denotes significant differences between groups (*p* < 0.01).

**Figure 2 children-10-00053-f002:**
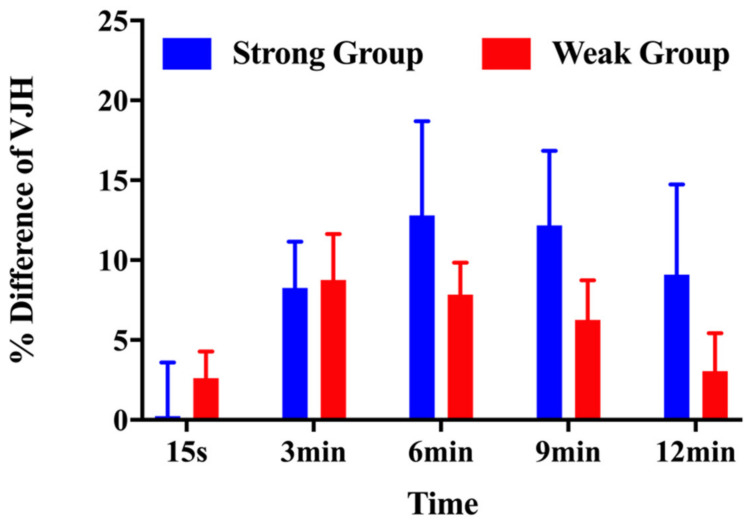
Percentage differences in the vertical jump height for the strong and weak groups in response to the conditioning activity at each time interval.

**Figure 3 children-10-00053-f003:**
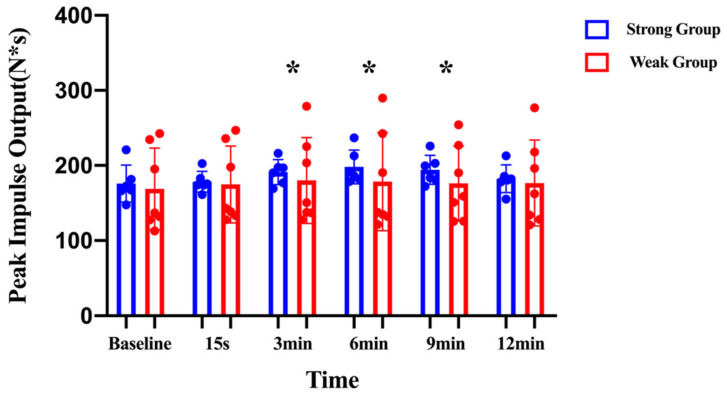
Peak impulses before and after the conditioning activity for the strong and weak groups. * denotes significant differences between groups (*p* < 0.05).

**Figure 4 children-10-00053-f004:**
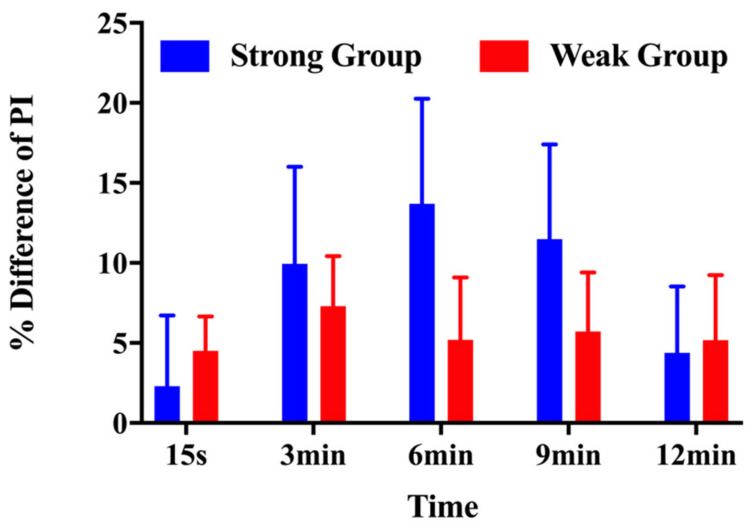
Percentage differences in peak impulse for the strong and weak groups in response to the conditioning activity at each time interval.

**Figure 5 children-10-00053-f005:**
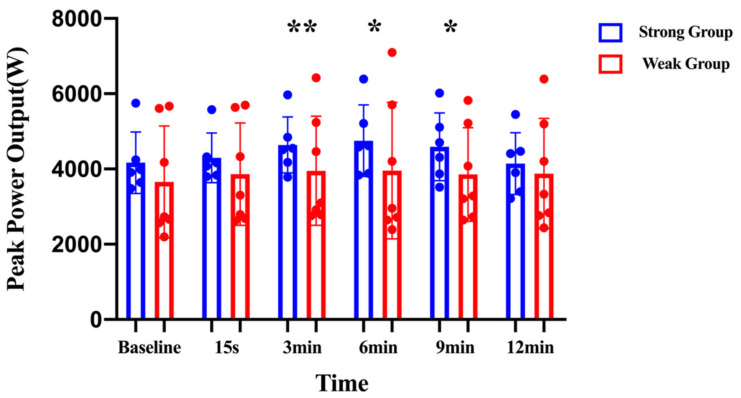
Peak power before and after the conditioning activity for the strong and weak groups. * denotes significant differences between groups (*p* < 0.05); ** denotes significant differences between groups (*p* < 0.01).

**Figure 6 children-10-00053-f006:**
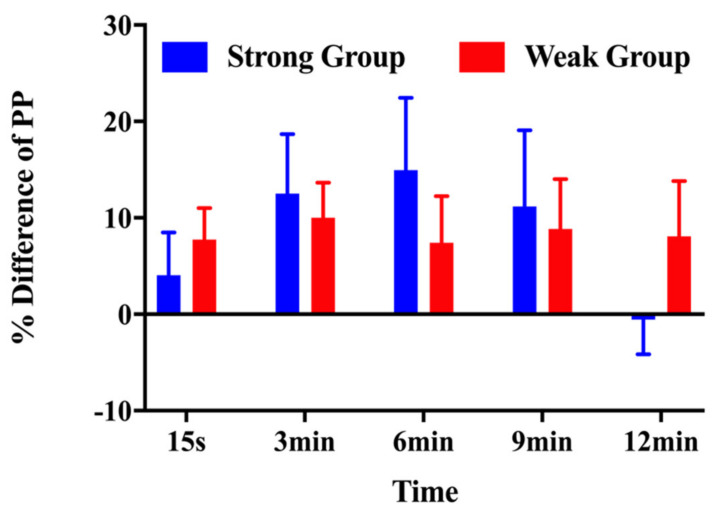
Percentage differences in peak power for the strong and weak groups in response to the conditioning activity at each time interval.

**Table 1 children-10-00053-t001:** Characteristics of the participants (*n* = 13).

Variable	Strong Group (*n* = 6) Mean ± *SD*	Weak Group (*n* = 7) Mean ± *SD*
Age (y)	19.7 ± 1.0	20.1 ± 1.4
Body mass (kg)	67.7 ± 5.5	70.7 ± 7.1
Height (cm)	175.7 ± 3.4	180.6 ± 3.8
Training (y)	6.7 ± 1.4	5.1 ± 1.1
1RM (kg)	188.3 ± 21.4	138.6 ± 6.9
1RM/BM	2.8 ± 0.3	2.0 ± 0.2

Note: 1RM, 1 repetition maximum; BM, body mass; *SD*, standard deviation.

## Data Availability

The data used to support the findings of current study are available from the corresponding author upon request.

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
