# Peer review of "Time Duration of Post-Activation Performance Enhancement (PAPE) in Elite Male Sprinters with Different Strength Levels"

_children, 2022, doi:10.3390/children10010053_

Round 1

Reviewer 1 Report

It is a very valuable study with good methodology, and great actuality and importance, for that congratulations, i find only  three (3) difficulties:

1.  The N considered for the groups are  small, and it is not statistically supported how this value was found

2. It would be good to provide the reference to consider the group with  IRF > 2.5 strong or not, if this information is available it should be included

3. Support the choice of times used for the study: 15s, 3-minute, 6-minute, 9-minute and 12-minute following a conditioning activity

Author Response

Dear Reviewer, 

I am attaching the changes.

Best regards, 

Reviewer 2 Report

Overall, the manuscript is well written and presented. The topic addressed is interesting and has been of great interest in the context of sports performance. 

I don't have many comments or corrections, however, I highlight two methodological issues that somehow compromise the quality of the study:

1. There is no control group or condition. A crossover design would have easily resolved this issue.

2. While the idea is interesting, there are definitely not enough athletes evaluated to divide the group into two categories (stronger and weaker) and compare using inferential statistics. This comparison could be performed but using only descriptive statistics (effect size, for example).

Author Response

(The authors gave the same response as above.)

Round 2

Reviewer 2 Report

In addition to the small sample in each group, the authors should add as limitations of the study the lack of a control group or a crossover design. The fact that previous studies did not use a control group in this type of study cannot be a justification for not using a control group in your study, since this is considered a methodological weakness.

Author Response

Thank you!

Best regards, 
